# An overview of the characteristics and quality assessment criteria in systematic review of pharmacoeconomics

**Chen Min**[1,2], **Mi Xue**[1,2], **Fei Haotian**[1,2], **Li Jialian**[1,2], **Zhang Lingli**[1,2]*

1 Department of Pharmacy/Evidence-Based Pharmacy Center, West China Second University Hospital, Sichuan University, Chengdu, Sichuan, P. R. China, 2 Key Laboratory of Birth Defects and Related Diseases of Women and Children (Sichuan University), Ministry of Education, Chengdu, Sichuan, P. R. China

* zhanglingli@scu.edu.cn

**Data Availability Statement:** All relevant data are within the manuscript and its Supporting Information files.

**Funding:** This review was funded by the Health Commission of Sichuan Province to CM and FH

## Abstract

### Background

The systematic review of economic evaluations plays a critical role in making well-informed decisions about competing healthcare interventions. The quality of these systematic reviews varies due to the lack of internationally recognized methodological evaluation standards.

### Methods

Nine English and Chinese databases including the Cochrane Library, PubMed, EMbase (Ovid), NHS economic evaluation database (NHSEED) (Ovid), Health Technology Assessment (HTA) database, Chinese National Knowledge Infrastructure (CNKI), WangFang, VIP Chinese Science & Technology Periodicals (VIP) and Chinese Biomedical Literature Database (CBM) were searched. Two reviewers independently screened studies and extracted data. The methodological quality of the literature was measured with modified AMSTAR. Data were narrative synthesized.

### Results

165 systematic reviews were included. The overall methodological quality of the literature was moderate according to the AMSTAR scale. In these articles, thirteen quality assessment tools and 32 author self-defined criteria were used. The three most widely used tools were the Drummond checklist (19.4%), the BMJ checklist (15.8%), the Consolidated Health Economic Evaluation Reporting Standards (CHEERS) statement (12.7%). Others included the Quality of Health Economic Studies (QHES), the Consensus on Health Economic Criteria (CHEC), the checklist of Center for Reviews and Dissemination (CRD), the Philips checklist, the World Health Organization (WHO) checklist, the checklist of Critical Appraisal Skills Program (CASP), the Pediatric Quality Appraisal Questionnaire (PQAQ), the Joanna Briggs Institute (JBI) checklist, Spanish and Chinese guidelines. The quantitative scales used in these literature were the QHES and PQAQ.

(No. 20PJ069). The funder has no role in the study design, data collection and analysis, decision to publish or preparation of the manuscript.

**Competing interests:** The authors have declared that no competing interests exist.

## Conclusions

Evidence showed that pharmacoeconomic systematic reviews' methodology remained to be improved, and the quality assessment criteria were gradually unified. Multiple scales can be used in combination to evaluate the quality of economic research in different settings and types.

## Introduction

It has become a great challenge for health policy-makers concerning the issue of realizing a more efficient allocation of limited medical and health resources [1, 2]. As a branch of health economics, pharmacoeconomic evaluations have been widely applied for healthcare decision-making and are essential for health technology assessment (HTA) [3, 4]. However, with an increasing amount of economic evaluations, healthcare professionals, consumers and policy-makers can often be overwhelmed by the presence of different results [5, 6]. The systematic review (SR) of these studies is considered a useful approach in making well-informed decisions on competing healthcare interventions [7].

Unlike the opponents, advocates such as Cochrane Collaboration, National Institute for Health and Care Excellence (NICE), Pharmaceutical Benefits Advisory Committee (PBAC), Canadian Agency for Drugs and Technologies in Health (CADTH), believed that the results of a pharmacoeconomic SR could help in public health assessment and policy-making [5, 8–11]. They maintained that it was not to produce the authoritative result on the relative cost-effectiveness but rather help decision-makers understanding the structure and potential impact of resource allocation issues. It can identify for decision-makers the range and quality of available studies and gaps in the evidence base [12]. In order to maximize its usefulness, pharmacoeconomic SR should be conducted in a systematic process. However, their quality varies due to the lack of internationally recognized methodological evaluation standards and the paucity of detailed reports [13, 14]. In 2002, the Cochrane Economic Collaboration Group assessed the quality of 39 economic evaluation SRs published in 1990–2001, with a 6-item self-made checklist. The results showed that the quality was satisfactory, but the search strategies and instruments to assess included studies' quality needed to pay more attention [15]. The weaknesses were that the database was not thoroughly searched and the quality evaluation of included studies was not described in approximately half and one-third of literature, respectively. Another SR conducted by Luhnen et al. in 2018 concluded that the methodologies applied for 83 SRs of health economic evaluations in HTA agencies and their reporting quality were very heterogeneous [16]. Process steps of these SRs such as data extraction, methodology quality or applicability assessment were frequently not performed at all, not reported or reported nontransparently.

Appraising the quality of included individual studies is important for performing pharmacoeconomic SRs. Using a well-developed checklist will make pharmacoeconomic SRs more transparent, informative and comparable [6]. A number of international organizations and researchers such as the British Medical Journal (BMJ), the International Society for Pharmacoeconomics and Outcomes Research (ISPOR) have developed generic and specific guidelines, checklists, and recommendations to conduct and report acceptable methods of economic analysis [17, 18]. Therefore, this study aimed to overview, summarize, and analyze pharmacoeconomic SRs' characteristics and the quality assessment tool used for the included individual

studies. It would be useful for future researchers wanting to conduct a SR of evidence about economic evaluations.

## Materials and methods

### Search strategy

The computer-based retrieval was conducted in the Cochrane Library, PubMed, EMbase (Ovid), NHS economic evaluation database (NHSEED) (Ovid), HTA Database, Chinese National Knowledge Infrastructure (CNKI), WanFang Data, VIP Chinese Science & Technology Periodicals (VIP) and Chinese Biomedical Literature Database (CBM) on October 25, 2019. Reference lists of relevant published studies and grey literature were also searched. The combination of MeSH terms and free text words were used for retrieval, with the corresponding adjustment according to the specific database. English search terms were "cost-benefit analysis", "cost-utility analysis", "cost-effectiveness analysis", "cost analysis economic evaluation", "review literature" and "systematic review". Meanwhile, Chinese search terms were "economics", "cost-effectiveness", "cost-benefit", "cost-utility" and "systematic review". The specific search strategies were shown in the S1 Text.

### Selection criteria

We used PICOS in defining explicit inclusion and exclusion criteria.

Inclusion criteria: (1) Patients: individuals with identified disease category; (2) Interventions/Comparators: informed drug intervention for disease prevention, treatment, diagnosis, etc.; and (3) Outcome/Setting: SRs of economic evaluations or HTA including SRs of economic evaluations.

Exclusion criteria: (1) literature published in other languages other than Chinese and English; (2) duplicate publication; (3) SRs of economic evaluations related to the discussion of disease burdens; (4) literature without access to the full text, such as conference summary; (5) quality evaluation or general review that was not a SR; and (6) SR on the comparison of drug and non-drug interventions.

### Study selection and data extraction

Two researchers independently completed the literature screening and data extraction according to the pre-determined inclusion and exclusion criteria, and the results were crossed checked. In case of disagreement, a discussion was made for clarification or resolved through a third reviewer. Data extraction consisted of general information (including author, year of publication, country of publication, journal of publication, quality appraisal tool, result synthesis method) and essential characteristics of quality appraisal criteria (including name, number of items, categories, etc).

### Study quality assessment

We used the Measure Tool to Assess Systematic Review (AMSTAR), an instrument for assessing the quality of SR of health care interventions, to evaluate the quality of included SRs [19]. The modified AMSTAR scale has the following ten items: (1) Was an "a priori" design provided? (2) Was there a duplicate study selection and data extraction? (3) Was a comprehensive literature search performed? (4) Was the status of publication (i.e. grey literature) used as an inclusion criterion? (5) Was a list of studies (included and excluded) provided? (6) Were the characteristics of the included studies provided? (7) Was the scientific quality of the included studies assessed and documented? (8) Was the scientific quality of the included studies used

appropriately in formulating conclusions? (9) Were the methods used to combine the findings of studies appropriate? (10) Was the conflict of interest included? Each questions was answered with "yes" (score 1), "no" (score 0) and "impossible to judge" (score 0.5). The final score was classified as a low score (< 5), medium score (5–8), or high score (>8). In this overview, item 7 of the AMSTAR scale refers to using checklists or guidelines to assess the quality of included economic studies.

## Data analysis

Microsoft Excel 2013 software was used to input the data and analyze the essential characteristics of the included literature, including the types of journals published, disease categories, quality evaluation methods of the included research, etc. A qualitative description was made on the quality assessment tool used for the included individual studies in pharmacoeconomic SR. We provided numbers and percentages for nominal data and median and ranges for count data.

# Results

## Study search and selection

A total of 11, 172 articles in English, 1, 004 articles in Chinese from databases and 17 articles obtained through google scholar were obtained in the initial literature retrieval. 9,938 articles were obtained after excluding duplicate publications. Of the titles and abstracts screened, 2,524 were ordered as full papers and assessed in detail. After independent screening by two researchers, 148 articles in English and 17 articles in Chinese were finally included for the overview. The process and results of the literature screening were shown in Fig 1. The references of 165 included articles were listed in the S2 Text.

## Characteristics of the included studies

The number of pharmacoeconomic SR literature publications gradually increased in the past 20 years (Fig 2). Of the 165 studies, More than two-thirds of articles were conducted in the United Kingdom, China, Canada, Netherlands, and the United States. Approximately half of the articles were published in professional economics journals such as HTA and Pharmacoeconomics. 3.0% of the full articles were published as a dissertation. Neoplasms were the most focused disease in these studies, accounting for 20.6%, followed by some infectious and parasitic diseases (18.8%). Furthermore, 19 disease categories (classified by the International Classification of Diseases-10) were involved. In these SRs, the most commonly searched databases included general databases such as MEDLINE, Embase, the Cochrane Library and the specialized databases of economic evaluations such as NHSEED and HTA database. 144 studies (87.3%) explicitly stated an assessment of the quality of included economic studies with a checklist or guideline recommendations, and 14 of which combined different tools. Data synthesis for all articles was qualitative descriptive analysis. The characteristics of 165 articles were shown in Table 1.

## Study quality assessment

According to the modified AMSTAR scale's evaluation results, the overall methodological quality was moderate, and the score ranged from 1 to 10, with an average score of 6.3. Of the included 165 literature, 36 were of low quality, 89 were of medium quality and 40 were of high quality. Three of the 10 quality items (comprehensive literature search, characteristics of the included studies, quality assessment) were fulfilled in at least 85% of reviews, with 88.5% (item

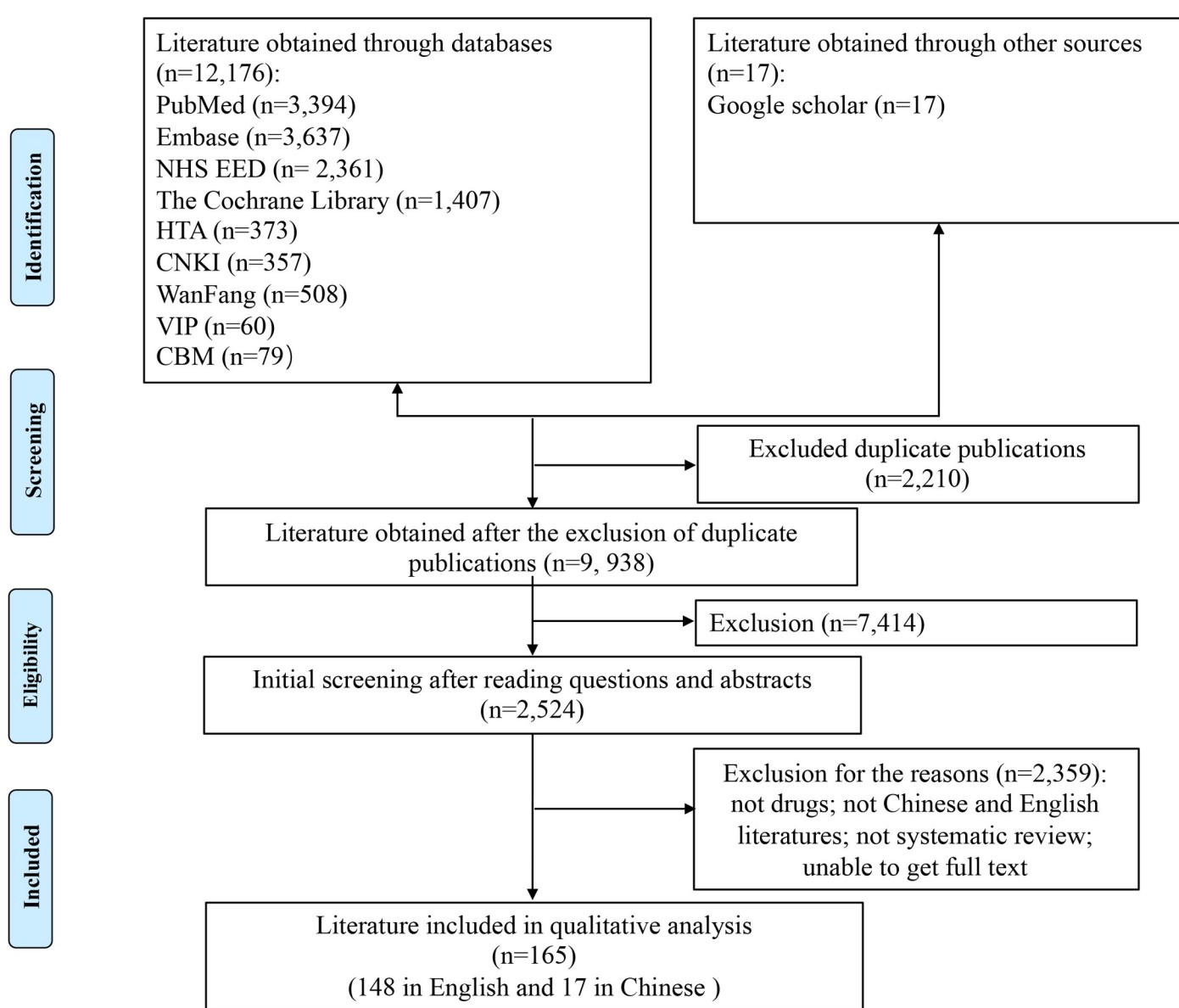

**Fig 1. PRISMA flow chart of literature search.** This flow diagram illustrated the search results and the process of screening and selecting studies for inclusion, and the reasons for exclusions in this review.

3), 94.5% (item 6) and 90.9% (item 7), respectively. However, 2 items (priori design and result synthesis) were fulfilled just about or less than thirty percent, with 30.9% (item 1) and 20.6% (item 9), respectively. A detailed methodological quality assessment of the pharmacoeconomic SR was shown in Fig 3 and S1 Table.

## Quality assessment tool used for the included studies

Table 2 detailed the characteristics of the checklists or guidelines. Eleven published checklists and two quality appraisal tools recommended by guidelines were used in the included 165 pharmacoeconomic SRs. And tools as follows had a relatively high frequency of use, including the Drummond checklist (19.4%), the BMJ checklist (15.8%), the Consolidated

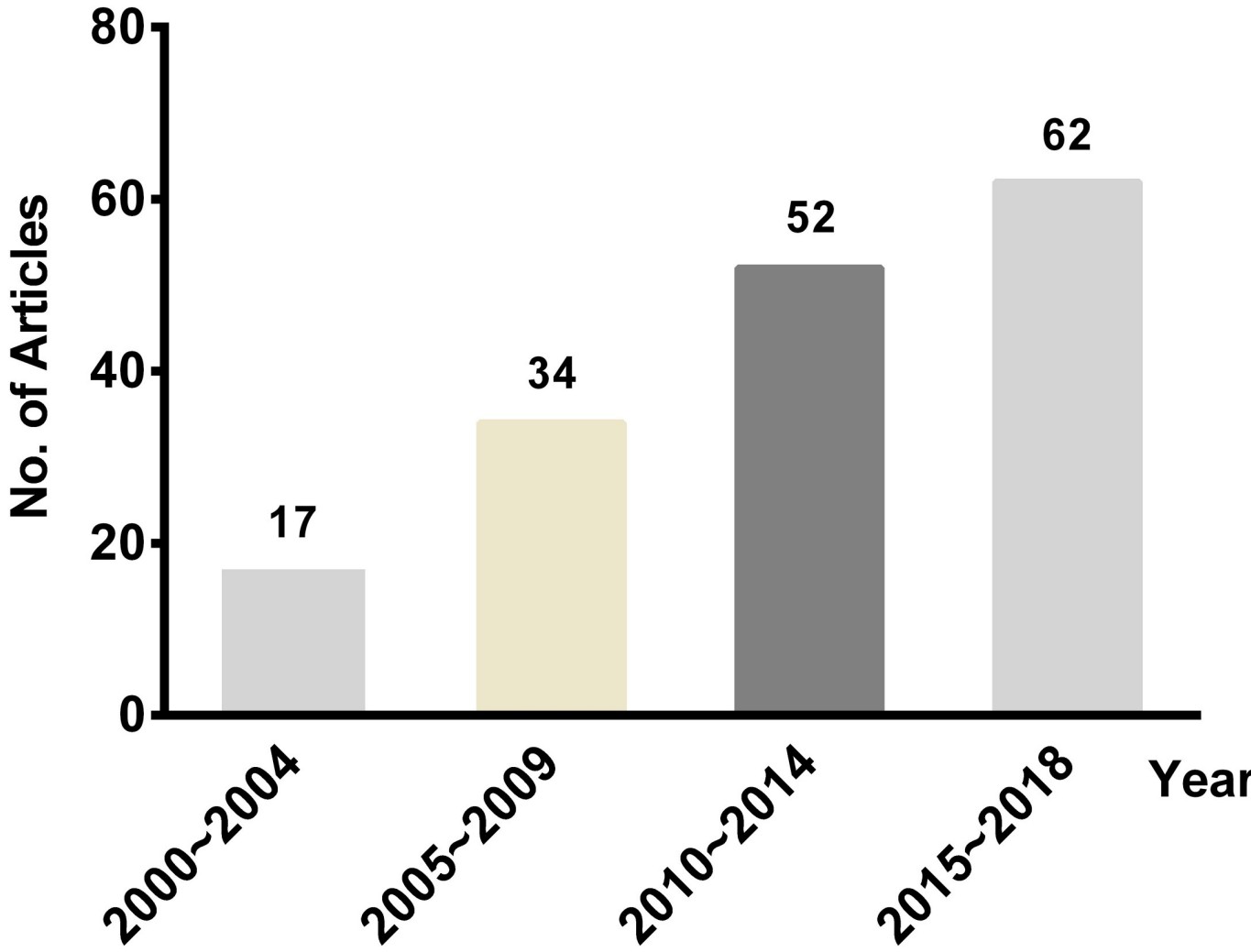

**Fig 2. The number of published pharmacoeconomic systematic reviews, 2000–2018.** This plot listed the number of pharmacoeconomic SR literature published in different years. The number on top of the bars represented the number of studies.

Health Economic Evaluation Reporting Standards (CHEERS) statement (12.7%). Besides, 32 studies modified the already published quality evaluation tools, such as the Drummond Checklist, the BMJ Checklist, the Consensus on Health Economic Criteria (CHEC) list, the CHEERS list and the Philips Checklist that were frequently applied. 21 articles did not evaluate the quality of the included studies. Among the checklists, the number of items ranged from 7 for model-based economic evaluation in the Drummond et al. Checklist to 57 for trial-and model-based economic evaluation in the Pediatric Quality Appraisal Questionnaire (PQAQ). Compared with the generic scale, the Quality of Health Economic Studies (QHES), the Philips and Joanna Briggs Institute (JBI) checklists were created to analyze the quality of economic evaluations based on modeling studies and the CHEC checklist on trial-based studies. Three articles used a disease-specific checklist created by the World Health Organization (WHO) to appraise the quality of the immunization program's economic evaluations, such as vaccines. While the majority of instruments contain personal and open-ended items, the QHES and PQAQ checklists provide a score to enable a simple comparison among studies.

**Table 1. Characteristics of the included 165 systematic reviews.**

| Category | Characteristic | Article (n = 165) | Percentage (%) |
|---|---|---|---|
| **Country** | United Kingdom | 56 | 33.9 |
| | China | 21 | 12.7 |
| | Canada | 15 | 9.1 |
| | Netherlands | 12 | 7.3 |
| | United States | 11 | 6.7 |
| | Others | 50 | 30.3 |
| **Journal** | Health Technology Assessment | 52 | 31.5 |
| | Pharmacoeconomics | 24 | 14.5 |
| | Expert Review of Pharmacoeconomics & Outcomes Research | 5 | 3.0 |
| | Vaccine | 5 | 3.0 |
| | Human Vaccines & Immunotherapeutics | 4 | 2.4 |
| | PLOS ONE | 4 | 2.4 |
| | China Pharmacy | 3 | 1.8 |
| | Others | 68 | 41.2 |
| **Disease** | Neoplasms | 34 | 20.6 |
| | Certain infections and parasitic diseases | 31 | 18.8 |
| | Disease of the circulatory system | 15 | 9.1 |
| | Disease of the respiratory system | 13 | 7.9 |
| | Diseases of the musculoskeletal system and connective tissue | 12 | 7.3 |
| | Endocrine, nutritional and metabolic diseases | 12 | 7.3 |
| | Others | 48 | 29 |
| **Intervention** | Anti-tumor medication | 31 | 18.8 |
| | vaccines | 21 | 12.7 |
| | Anti-infective medication | 19 | 11.5 |
| | Endocrine system medication | 17 | 10.3 |
| | Immunosuppressive medication | 17 | 10.3 |
| | Blood and hematopoietic system medication | 12 | 7.3 |
| | Central system medication | 12 | 7.3 |
| | Others | 36 | 21.8 |
| **Search Database** | MEDLINE | 120 | 72.7 |
| | EMbase | 103 | 62.4 |
| | The Cochrane Library | 92 | 55.8 |
| | NHS economic evaluation database | 79 | 47.9 |
| | Health Technology Assessment database | 62 | 37.6 |
| **Quality assessment tool** | Customized | 32 | 19.4 |
| | Drummond | 32 | 19.4 |
| | BMJ | 26 | 15.8 |
| | CHEERS | 21 | 12.7 |
| | None | 21 | 12.7 |
| | QHES | 10 | 6.1 |
| | CHEC | 7 | 4.2 |
| | CRD | 5 | 3.0 |
| | Philips | 3 | 1.8 |
| | WHO | 3 | 1.8 |
| | CASP | 1 | 0.6 |
| | PQAQ | 1 | 0.6 |
| | JBI | 1 | 0.6 |
| | Spanish guideline | 1 | 0.6 |
| | Chinese guideline | 1 | 0.6 |

*(Continued)*

**Table 1.** (Continued)

| Category | Characteristic | Article (n = 165) | Percentage (%) |
|---|---|---|---|
| Data synthesis | Narrative synthesis | 165 | 100 |

**BMJ,** British Medical Journal; **CHEERS,** Consolidated Health Economic Evaluation Reporting Standards; **QHES,** Quality of Health Economic Studies; **CRD,** Center for Reviews and Dissemination; **CHEC,** Consensus on Health Economic Criteria; **CASP,** Critical Appraisal Skills Program; **WHO,** World Health Organization; **PQAQ,** Pediatric Quality Appraisal Questionnaire; **JBI,** Joanna Briggs Institute.

## Discussion

Our review's findings indicated that the amount of pharmacoeconomic SR published has grown gradually from 2000 to 2018. Unlike the literature of Jefferson et al. [15] and Luhnen et al. [16], we evaluated the quality of the latest 165 SRs and searched for more comprehensive databases. Our review results revealed that the two methodological flaws described by Jefferson (unsatisfied search strategies and quality assessment of included studies) appeared to have improved but still needed further refinement. In our research, the majority of 165 articles (78.2%) had medium or high methodological quality according to the modified AMSTAR scale. Most of the SRs were reported following the Preferred Reporting Items for Systematic Reviews and Meta-Analyses (PRISMA) procedures, but two parts were worth noting.

First, there was no preliminary study design, with only 51 literature (30.9%) providing registration numbers. It is recommended to write a protocol for the SR of economic evaluations in advance. Second, it is difficult to answer whether the method of pooling the research results was appropriate. Identified economic evaluations are usually too heterogeneous to conduct a meta-analysis of results [12]. This is in line with our findings that all review's results were synthesized narratively and 94.5% of the SRs provided the characteristics of the included studies. It may be useful to include summary tables that present key information relating to costs and consequences, including but not limited to population, country, perspective, comparison of interventions, the measurement of effectiveness and incremental cost-effectiveness ratio (ICER) [9]. It is suggested that the focus of pharmacoeconomic SRs is not to try to generate aggregate estimates of cost-effectiveness ratios but rather to explain the reasons for this

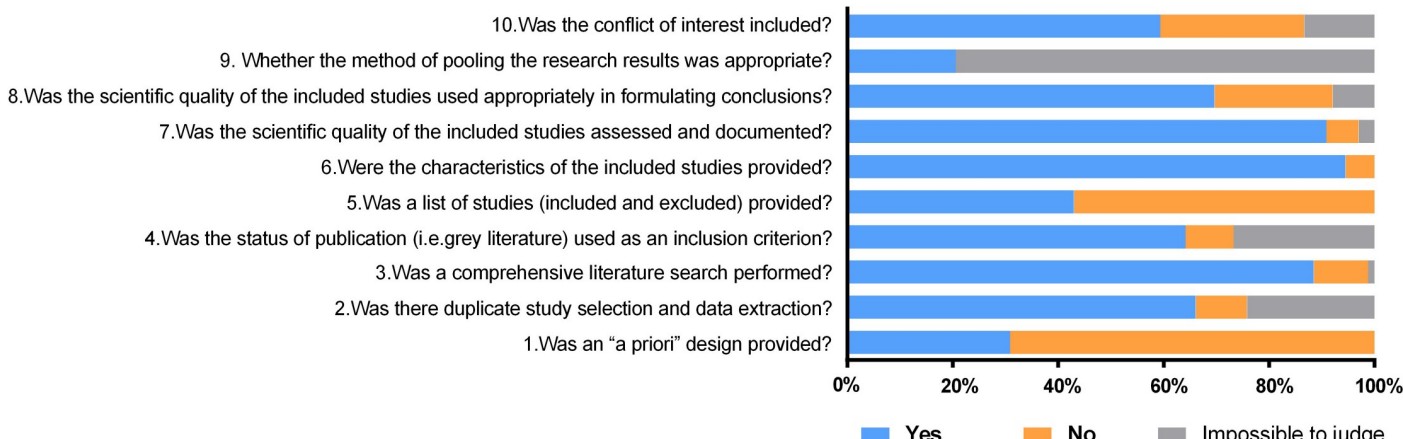

**Fig 3. The methodological quality was evaluated by the modified AMSTAR scale in the pharmacoeconomic systematic review.** This plot illustrated the quality of included studies with each judgment ("yes" (score 1), "no" (score 0) and "impossible to judge" (score 0.5)). The overall methodological quality was moderate. Three of the 10 quality items (item 3, item 6 and item 7) were fulfilled in at least 85% of reviews, while 2 items (item 1 and item 9) were fulfilled just about or less than thirty percent.

**Table 2. Summary of the existing checklist used in the pharmacoeconomic systematic review.**

| Author,Year | Tools | Country | Journal | Setter | Items | Economic Evaluation Type | Response | Scored | Percentage (%) |
|---|---|---|---|---|---|---|---|---|---|
| Drummond 2005 [20] | Drummond checklist | UK | International Journal of Technology Assessment in HealthCare | - | 10 (Trial) 7 (model) | Trial-and model-based | Yes, no, can't tell | No | 19.4 |
| Drummond 1996 [17] | BMJ checklist | UK | BMJ | BMJ Economic Evaluation Party | 35 | Trial-and model-based | Yes, no, not clear, NA | No | 15.8 |
| Husereau 2013 [18] | CHEERS | Canada | BMJ | ISPOR | 24 | Trial-and model-based | Yes, no | No | 12.7 |
| Chiou 2003 [21] | QHES | USA | Medical Care | - | 16 | Model-based | Yes, no | Yes | 6.1 |
| Evers 2005 [6] | CHEC | Netherlands | International Journal of Technology Assessment in Health Care | - | 19 | Trial-based | Yes, no | No | 4.2 |
| CRD 2009 [9] | CRD | UK | Online | CRD University of York | 36 | Trial-and model-based | Yes, no | No | 3.0 |
| Philips 2004 [22] | Philips | UK | pharmacoeconomics | - | 60 | Model-based | Yes, no, unclear, NA | No | 1.8 |
| Walker 2010 [23] | WHO checklist | Switzerland | Vaccine | Department of Immunization, Vaccines and Biologicals | 43 | Vaccine-specific | Yes, no, partially, not clear, NA | No | 1.8 |
| Ungar 2003 [24] | PQAQ | Canada | Value in Health | - | 57 | Trial-and model-based pediatric-specific | Yes, no, NA | Yes | 0.6 |
| López 2010 [25] | Spanish guideline | spain | European Journal of Health Economics | - | - | Trial-and model-based | Yes, no, in part, NA | No | 0.6 |
| Liu 2011 [26] | Chinese guideline | China | China Journal of Pharmaceutical Economics | Pharmacoeconomic Evaluations Group | - | Trial-and model-based | Yes, no | No | 0.6 |
| CASP 2013 [27] | CASP | UK | Online | Oxford Centre for Triple Value Healthcare | 12 | Trial-and model-based | yes, no, can't tell | No | 0.6 |
| Gomersall 2015 [12] | JBI | Australia | International Journal of Evidence-Based Healthcare | Joanna Briggs Institute | 11 | Model-based | Yes, no, unclear, NA | No | 0.6 |

**BMJ**, British Medical Journal; **CHEERS**, Consolidated Health Economic Evaluation Reporting Standards; **QHES**, Quality of Health Economic Studies; **CRD**, Center for Reviews and Dissemination; **CHEC**, Consensus on Health Economic Criteria; **CASP**, Critical Appraisal Skills Program; **JBI**, Joanna Briggs Institute; **WHO**, World Health Organization; **PQAQ**, Pediatric Quality Appraisal Questionnaire; **ISPOR**, International Society for Pharmacoeconomics and Outcomes Research. "-", not available; "NA", not appropriate.

difference from setting to setting. This is also an approach to improve the generalizability and transferability of data.

There were several checklists/guidelines for guiding researchers in conducting and reporting acceptable economic analysis. Walker et al. [28] reported ten original checklists for assessing an economic evaluation from 1992 to 2011. Zhang et al. [29] reported twelve original checklists with good reliability and validity from 1987 to 2013. In this overview, we retrieved 13 critical appraisal tools developed by institutions and several customized scales based on these pharmacoeconomic evaluation tools. The Drummond checklist, the BMJ checklist, and the CHEERS scale were the three-most widely used quality assessment tools in pharmacoeconomic SR. The Drummond checklist was from the monograph "Methods for the Economic Evaluation of Health Care Programmes", proposed by professor Drummond in 1987. Based on

the successive four consecutive updates, it comprehensively and systematically introduced the principles, concepts, methods, and economic evaluation applications. It has a wide range of global influence and is an essential guideline for economic evaluation in the healthcare field [30]. The BMJ Economic Evaluation Working Party published the BMJ checklist in 1996 to improve the quality of submitted and published economic articles, which played an important role in economic research [17]. In 2013, through a 2-rounds Delphi technique, the ISPOR developed a 24 items CHEERS checklist [18]. Customized evaluation scales were used in 32 literature. For example, a pharmacoeconomic SR published in Lancet in 2002, which evaluated the cost-effectiveness of various interventions for HIV patients, customized the methods of literature quality appraisal with eight evaluation items [31]. In our previous study of evaluating therapeutic drugs for treating immune thrombocytopenia, the evaluation method was also developed based on the Drummond list and the CHEC scale, with 13 evaluation items customized [32]. Another paper, published in HTA, concerning the pharmacoeconomic SR of chronic myelogenous leukemia, referred to the Drummond list and defined 18 items to evaluate the quality of the original economic research [33].

Although the use of checklists/guidelines will not guarantee that the results of an economic SR are valid, it can ensure that the review has appropriate components [28]. In our view, the Drummond checklist, the BMJ checklist, and the CHEERS scale could be adopted as a quality assessment tool to assess the methodological quality of full and partial economic evaluations, and the Philips checklist especially for modeling evaluations. It should be noted that the above quality assessment tools are used for qualitative evaluation, which contain personal and open-ended items. In the case of quantitative assessment, QHES and PQAQ scales are recommended. In addition to generic ones, there are population-specific scales (such as PQAQ) and disease-specific scales (such as the WHO vaccine checklist). Finally, different scales may address different types of economic assessments, such as trial-based and model-based. Therefore, different studies may combine multiple different scales.

There were several limitations to our approach. Firstly, our overview did not include literature published other than Chinese and English, leading to publication bias. Secondly, the literature search was updated on October 25, 2019. The number of pharmacoeconomic SR has indeed increased, possibly leading to changes in results. We believe these problems will not have a significant impact on our findings. Thirdly, we did not conduct a statistical analysis comparing the quality of English versus Chinese, with or without checklists/guidelines. Because AMSTAR is not designed to generate an overall "score". A high score on total items may disguise critical weaknesses in specific domains, such as an inadequate literature search or failure to assess the risk of bias with individual studies included in a systematic review. We also did not conduct a statistical analysis comparing the quality of different checklists/guidelines, owing to the lack of standardization for the quality assessment of economic evaluation. Each of these quality assessment tools has its characteristics and applicable conditions.

## Conclusion

Our results identified a number of well-developed quality assessment criteria available to researchers to ensure the informative and transparent SR of economic assessments. The Drummond checklist, the BMJ checklist, and the CHEERS criteria were the three-most widely used tools. Evidence showed that the methodology of pharmacoeconomic SRs remained to be improved according to the AMSTAR scale. Considering the different types and population of included studies, a multiple scales may need to be combined in future pharmacoeconomic SRs.

## Supporting information

**S1 Text. Search strategies for English and Chinese databases.**
(DOCX)

**S2 Text. The references of included 165 studies.**
(DOCX)

**S1 Table. AMSTAR assessment scale for included 165 studies.**
(DOCX)

**S1 Checklist. PRISMA checklist.**
(DOC)

## Author Contributions

**Data curation:** Mi Xue, Fei Haotian, Li Jialian.

**Formal analysis:** Mi Xue, Fei Haotian, Li Jialian.

**Methodology:** Chen Min, Zhang Lingli.

**Supervision:** Zhang Lingli.

**Writing – original draft:** Chen Min, Mi Xue, Fei Haotian, Li Jialian, Zhang Lingli.

**Writing – review & editing:** Chen Min, Zhang Lingli.

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
