## [Decision Letter · Decision Letter 0]

13 Oct 2020

PONE-D-20-23142

An Overview of the Characteristics and Quality Assessment Criteria in Systematic Review of Pharmacoeconomics

PLOS ONE

Dear Dr. chen,

Thank you for submitting your manuscript to PLOS ONE. After careful consideration, we feel that it has merit but does not fully meet PLOS ONE’s publication criteria as it currently stands. Therefore, we invite you to submit a revised version of the manuscript that addresses the points raised during the review process.

We look forward to receiving your revised manuscript.

Kind regards,

Kevin Lu, PhD

Academic Editor

PLOS ONE

Journal Requirements:

Reviewers' comments:

Reviewer's Responses to Questions

**Comments to the Author**

1. Is the manuscript technically sound, and do the data support the conclusions?

Reviewer #1: Partly

Reviewer #2: Yes

2. Has the statistical analysis been performed appropriately and rigorously? 

Reviewer #1: Yes

Reviewer #2: No

3. Have the authors made all data underlying the findings in their manuscript fully available?

Reviewer #1: No

Reviewer #2: Yes

4. Is the manuscript presented in an intelligible fashion and written in standard English?

Reviewer #1: Yes

Reviewer #2: Yes

5. Review Comments to the Author

Reviewer #1: Comment #1: Please provide the complete search terms and search results for each database in appendix.

Comment #2: Please list specifically how many records were identified out in each database in the flowchart in Figure 1.

Comment #3: The review searched literatures in both English and Chinese, but there is no description of how many of the literatures included were in Chinese and how many were in English. Please add relevant information.

Comment #4, page 8: Please add references for the following statement.

1. “However, with an increasing amount of economic evaluations, healthcare professionals, consumers and policy-makers can often be overwhelmed by the presence of different results.”

2. “Opponents held that they were worthless because of differences between the populations, settings and background of the included studies.”

Comment #5, page 9: The literature search was up to October 25, 2019, almost a year before. Could the authors consider updating the literature search?

Comment #6, page 19: The PubMed search strategy provided in the manuscript seems a bit weird. Why was the term “cost-benefit analysis” searched in full text alone in #1 and again in the title and abstract in #2?

Comment #7, page 19: Please add information on the interventions of included studies in Table 1.

Reviewer #2: Major comments:

1 Introduction. Need to literature gap.

2 Methods. Need to distinguish between 2 types of tools: AMSTAR for systematic review and checklists/guidelines for included studies

3 Recommend to conduct statistical analysis comparing the quality of 1) English vs. Chinese, 2) with and without checklists/guidelines, 3) different checklists/guidelines, and other as you see fit.

4 Limitation. Need to develop.

5 How do you link AMSTAR for systematic review and checklists/guidelines for included studies? Do you recommend using checklists/guidelines to improve the quality of systematic review?

Minor comments:

1 Table 2. Add the classification of each tool. E.g. score range for high, medium, and low quality

2 Table 2. What does the column “Answer” mean?

3 Figure 1: What are other sources?

6. PLOS authors have the option to publish the peer review history of their article (what does this mean?). If published, this will include your full peer review and any attached files.

Reviewer #1: No

Reviewer #2: No

---

## [Author Response · Author response to Decision Letter 0]

26 Nov 2020

Journal Requirements:

 Response: Many thanks for these useful and valid suggestions!

We have revised the format.

 Response: Many thanks for these useful and valid suggestions!

We have added the supporting information.

Review Comments to the Author

Reviewer #1: Comment #1: Please provide the complete search terms and search results for each database in appendix.

Response: Many thanks for these useful and valid suggestions! 

We have listed the retrieval strategies for all English and Chinese databases in the supporting information. We have modified the sentences in the text “ The specific search strategies were shown in the S1 Text”.

Comment #2: Please list specifically how many records were identified out in each database in the flowchart in Figure 1.

Response: Many thanks for these useful and valid suggestions! 

We have listed the search records for each database in the flowchart in Figure 1. 

Comment #3: The review searched literatures in both English and Chinese, but there is no description of how many of the literatures included were in Chinese and how many were in English. Please add relevant information.

Response: Many thanks for these useful and valid suggestions! 

We have revised the sentences in the “ Results-Study search and selection” part as: “ A total of 11, 172 articles in English, 1, 004 articles in Chinese from databases and 17 articles obtained through google scholar were obtained in the initial literature retrieval.” and “ 148 articles in English and 17 articles in Chinese were finally included for the overview”.

Comment #4, page 8: Please add references for the following statement.

1. “ However, with an increasing amount of economic evaluations, healthcare professionals, consumers and policy-makers can often be overwhelmed by the presence of different results.”

2. “ Opponents held that they were worthless because of differences between the populations, settings and background of the included studies.”

Response: Many thanks for these useful and valid suggestions! 

We have added references:

1. Shemilt I, Mugford M, Byford S, Drummond M, Eisenstein E, Knapp M, et al. 15 Incorporating economics evidence. Cochrane handbook for systematic reviews of interventions. 2008:449. 

2. Evers S, Goossens M, De Vet H, Van Tulder M, Ament A. Criteria list for assessment of methodological quality of economic evaluations: Consensus on Health Economic Criteria. International journal of technology assessment in health care. 2005; 21(2):240-245. 

3. Anderson R. Systematic reviews of economic evaluations: utility or futility? Health Econ. 2010 Mar;19(3):350-64. 

Comment #5, page 9: The literature search was up to October 25, 2019, almost a year before. Could the authors consider updating the literature search?

Response: Many thanks for these useful and valid suggestions! 

We also considered retrieving the databases and updating to the latest date. After a rough search we found that the number of new system reviews did increase as of 2019 and 2020, but the evaluation tools used were similar. So we didn't update them. We have considered this problem a limitation in the “ discussion” part: “ Second, the literature search was updated on October 25, 2019. The number of pharmacoeconomic systematic review has indeed increased, possibly leading to changes in results. We believe these problems will not have a significant impact on our findings.”.

Comment #6, page 19: The PubMed search strategy provided in the manuscript seems a bit weird. Why was the term “ cost-benefit analysis” searched in full text alone in #1 and again in the title and abstract in #2?

Response: Many thanks for these useful and valid suggestions! 

It should be the mesh term “ cost-benefit analysis” in #1 and again in the title and abstract in #2. We have revised the search strategy. The research strategies for all English and Chinese databases were listed in the supporting information.

Comment #7, page 19: Please add information on the interventions of included studies in Table 1.

Response: Many thanks for these useful and valid suggestions! 

We have added information on the interventions of included studies in Table 1. 

Reviewer #2: Major comments:

1 Introduction. Need to literature gap.

Response: Many thanks for these useful and valid suggestions! 

We have revised the sentences and added some literature such as two existed systematic reviews (Jefferson 2002 and Luhnen 2018). For more modifications, please see the Introduction section.

2 Methods. Need to distinguish between 2 types of tools: AMSTAR for systematic review and checklists/guidelines for included studies.

Response: Many thanks for these useful and valid suggestions! 

AMSTAR is used for assessing the quality of systematic reviews and checklists/guidelines for included individual studies. We have added the ten items in the “ Methods- Study Quality Assessment” section as “ There are the following ten items: (1) Was an 'a priori' design provided? (2) Was there a duplicate study selection and data extraction? (3)Was a comprehensive literature search performed? (4) Was the status of publication (i.e. grey literature) used as an inclusion criterion? (5) Was a list of studies (included and excluded) provided? (6) Were the characteristics of the included studies provided?(7) Was the scientific quality of the included studies assessed and documented?(8) Was the scientific quality of the included studies used appropriately in formulating conclusions? (9) Were the methods used to combine the findings of studies appropriate? (10) Was the conflict of interest included?”

We have added “ In this overview, item 7 of the AMSTAR scale refers to using checklists or guidelines to assess the quality of included economic studies.”

3 Recommend to conduct statistical analysis comparing the quality of 1) English vs. Chinese, 2) with and without checklists/guidelines, 3) different checklists/guidelines, and other as you see fit.

Response: Many thanks for these useful and valid suggestions! 

It’s challenging to conduct a statistical analysis comparing these issues. We have added these issues in the “ discussion-limitation” section: “ Thirdly, we did not conduct a statistical analysis comparing the quality of English versus Chinese, with or without checklists/guidelines. Because AMSTAR is not designed to generate an overall “ score”. A high score on total items may disguise critical weaknesses in specific domains, such as an inadequate literature search or failure to assess the risk of bias with individual studies included in a systematic review. We also did not conduct a statistical analysis comparing the quality of different checklists/guidelines, owing to the lack of standardization for the quality assessment of economic evaluation. Each of these quality assessment tools has its characteristics and applicable conditions”.

4 Limitation. Need to develop.

Response: Many thanks for these useful and valid suggestions! 

We have revised the limitations: “ There are several limitations present in our approach. Firstly, our overview did not include literature published other than Chinese and English, which may lead to publication bias. Secondly, the literature search was updated on October 25, 2019. The number of pharmacoeconomic systematic review has indeed increased, possibly leading to changes in results. We believe these problems will not have a significant impact on our findings. Thirdly, we did not conduct a statistical analysis comparing the quality of English versus Chinese, with or without checklists/guidelines. Because AMSTAR is not designed to generate an overall “ score”. A high score on total items may disguise critical weaknesses in specific domains, such as an inadequate literature search or failure to assess the risk of bias with individual studies included in a systematic review. We also did not conduct a statistical analysis comparing the quality of different checklists/guidelines, owing to the lack of standardization for the quality assessment of economic evaluation. Each of these quality assessment tools has its characteristics and applicable conditions.”

5 How do you link AMSTAR for systematic review and checklists/guidelines for included studies? Do you recommend using checklists/guidelines to improve the quality of systematic review?

Response: Many thanks for these useful and valid suggestions! 

AMSTAR is used for assessing the quality of systematic reviews and checklists/guidelines for included individual studies. We have added “ In this overview, item 7 of the AMSTAR scale refers to using checklists or guidelines to assess the quality of included economic studies.” in the Method section.

As one of the essential items in the AMSTAR scale, we recommend using checklists/guidelines to improve the quality of systematic review. We have added the following sentences in the Discussion section: “ In our view, the Drummond checklist, the BMJ checklist, and the CHEERS scale could be adopted as a quality assessment tool to assess the methodological quality of full and partial economic evaluations, and Phillips checklist especially for modeling evaluations. It should be noted that the above quality assessment tools are qualitative evaluation, which contain personal and open-ended items. In the case of quantitative assessment, QHES and PQAQ scales are recommended. In addition to generic ones, there are population-specific scales (such as PQAQ) and disease-specific scales (such as the WHO vaccine checklist). Finally, different scales may address different types of economic assessments, such as trial-based and model-based. Therefore, different studies may combine multiple different scales.”.

Minor comments:

1 Table 2. Add the classification of each tool. E.g. score range for high, medium, and low quality

Response: Many thanks for these useful and valid suggestions! 

We detailed the characteristics of the checklists or guidelines in Table 2. It is hard to classify a score range of high, medium and low quality, because many checklists/guidelines do not provide a quantitative scoring criteria and rules. Usually those checklists/guidelines only provide some items to check whether a study contains specific contents. In the evaluation process, we suggest to customize the criteria of quality classification.

2 Table 2. What does the column “ Answer” mean?

Response: Many thanks for these useful and valid suggestions! 

It means some of the options for checklists/guidelines evaluation. We have revised the “ Answer” to “ Response”. 

 3 Figure 1: What are other sources?

Response: Many thanks for these useful and valid suggestions! 

“ Other sources” refer to the search results from “ Google scholar” in our overview. We have revised the sentences in the “ Study search and selection” section: “ A total of 11, 172 articles in English, 1, 004 articles in Chinese from databases and 17 articles obtained through google scholar were obtained in the initial literature retrieval.”.

---

## [Decision Letter · Decision Letter 1]

29 Dec 2020

PONE-D-20-23142R1

An overview of the characteristics and quality assessment criteria in systematic review of pharmacoeconomics

PLOS ONE

Dear Dr. chen,

Thank you for submitting your manuscript to PLOS ONE. After careful consideration, we feel that it has merit but does not fully meet PLOS ONE’s publication criteria as it currently stands. Therefore, we invite you to submit a revised version of the manuscript that addresses the points raised during the review process.

We look forward to receiving your revised manuscript.

Kind regards,

Kevin Lu, PhD

Academic Editor

PLOS ONE

Reviewers' comments:

Reviewer's Responses to Questions

**Comments to the Author**

1. If the authors have adequately addressed your comments raised in a previous round of review and you feel that this manuscript is now acceptable for publication, you may indicate that here to bypass the “Comments to the Author” section, enter your conflict of interest statement in the “Confidential to Editor” section, and submit your "Accept" recommendation.

Reviewer #1: All comments have been addressed

2. Is the manuscript technically sound, and do the data support the conclusions?

Reviewer #1: Yes

3. Has the statistical analysis been performed appropriately and rigorously? 

Reviewer #1: Yes

4. Have the authors made all data underlying the findings in their manuscript fully available?

Reviewer #1: Yes

5. Is the manuscript presented in an intelligible fashion and written in standard English?

Reviewer #1: Yes

6. Review Comments to the Author

Reviewer #1: Thanks to the authors for addressing the previous review comments well. However, there are some minor revisions that need to be made in the manuscript.

Comment 1: There are several grammatical errors or typos in the article, and I have listed some of them below. It would be great if the authors could proofread the article carefully and correct the errors.

1) Abstract background section: “A systematic review of economic evaluations plays a critical…..” to “The systematic reviews of economic evaluations plays a critical…..”

2) Abstract results section: “Others include the Quality of Health Economic Studies (QHES)” to “Others included the Quality of Health Economic Studies (QHES)”

3) Discussion: “…… these pharmcoeconomic evaluation tools.” to “…... these pharmacoeconomic evaluation tools.”

Comment 2: Please spell out the name of the database described in the methods section of the abstract.

7. PLOS authors have the option to publish the peer review history of their article (what does this mean?). If published, this will include your full peer review and any attached files.

Reviewer #1: No

---

## [Author Response · Author response to Decision Letter 1]

9 Jan 2021

Reviewer #1: Thanks to the authors for addressing the previous review comments well. However, there are some minor revisions that need to be made in the manuscript.

Comment 1: There are several grammatical errors or typos in the article, and I have listed some of them below. It would be great if the authors could proofread the article carefully and correct the errors.

1) Abstract background section: “A systematic review of economic evaluations plays a critical…..” to “The systematic reviews of economic evaluations plays a critical…..”

2) Abstract results section: “Others include the Quality of Health Economic Studies (QHES)” to “Others included the Quality of Health Economic Studies (QHES)”

3) Discussion: “…… these pharmcoeconomic evaluation tools.” to “…... these pharmacoeconomic evaluation tools.”

Response: Many thanks for these useful and valid suggestions! 

We have corrected the errors in the article. Please see the revision of the manuscript.

Comment 2: Please spell out the name of the database described in the methods section of the abstract.

Response: Many thanks for these useful and valid suggestions! 

We have detailed the name of the database described in the methods section of the abstract as following:

 “Nine English and Chinese databases including the Cochrane Library, PubMed, EMbase (Ovid), NHS economic evaluation database (NHSEED) (Ovid), Health Technology Assessment (HTA) database, Chinese National Knowledge Infrastructure (CNKI), WangFang, VIP Chinese Science & Technology Periodicals (VIP) and Chinese Biomedical Literature Database (CBM) were searched.”

---

## [Editor Report · Decision Letter 2]

13 Jan 2021

An overview of the characteristics and quality assessment criteria in systematic review of pharmacoeconomics

PONE-D-20-23142R2

Dear Dr. chen,

We’re pleased to inform you that your manuscript has been judged scientifically suitable for publication and will be formally accepted for publication once it meets all outstanding technical requirements.

Kind regards,

Kevin Lu, PhD

Academic Editor

PLOS ONE

---

## [Editor Report · Acceptance letter]

28 Jan 2021

PONE-D-20-23142R2 

An overview of the characteristics and quality assessment criteria in systematic review of pharmacoeconomics 

Dear Dr. Min:

I'm pleased to inform you that your manuscript has been deemed suitable for publication in PLOS ONE. Congratulations! Your manuscript is now with our production department. 

Kind regards, 

on behalf of

Professor Kevin Lu 

Academic Editor

PLOS ONE